# Direct observation of polymer surface mobility via nanoparticle vibrations

Hojin Kim[1], Yu Cang[2], Eunsoo Kang[2], Bartlomiej Graczykowski[2,3], Maria Secchi [4], Maurizio Montagna [5], Rodney D. Priestley[6], Eric M. Furst[1] & George Fytas[2,7]

Measuring polymer surface dynamics remains a formidable challenge of critical importance to applications ranging from pressure-sensitive adhesives to nanopatterning, where inter-facial mobility is key to performance. Here, we introduce a methodology of Brillouin light spectroscopy to reveal polymer surface mobility via nanoparticle vibrations. By measuring the temperature-dependent vibrational modes of polystyrene nanoparticles, we identify the glass-transition temperature and calculate the elastic modulus of individual nanoparticles as a function of particle size and chemistry. Evidence of surface mobility is inferred from the first observation of a softening temperature, where the temperature dependence of the fundamental vibrational frequency of the nanoparticles reverses slope below the glass-transition temperature. Beyond the fundamental vibrational modes given by the shape and elasticity of the nanoparticles, another mode, termed the interaction-induced mode, was found to be related to the active particle–particle adhesion and dependent on the thermal behavior of nanoparticles.

[1] Department of Chemical and Biomolecular Engineering, University of Delaware, Newark, DE 19716, USA. [2] Max Planck Institute for Polymer Research, Ackermannweg 10, 55128 Mainz, Germany. [3] NanoBioMedical Centre, Adam Mickiewicz University, ul. Umultowska 85, Poznan 61-614, Poland. [4] Department of Industrial Engineering, University of Trento, 38123 Trento, Italy. [5] Department of Physics, University of Trento, 38123 Trento, Italy. [6] Department of Chemical and Biological Engineering, Princeton University, Princeton, NJ 08544, USA. [7] IESL-FORTH, N. Plastira 100, 70013 Heraklion, Crete, Greece. Correspondence and requests for materials should be addressed to E.M.F. (email: furst@udel.edu) or to G.F. (email: fytas@mpip-mainz.mpg.de)

Polymer nanoparticles (NPs) are useful materials with applications in drug delivery[1–3], surface coatings[4,5], and assembly of photonic[6–8] and phonic materials[9,10]. Optimizing NPs for these applications requires an understanding of their physical properties, including the glass-transition temperature ($T_g$) and elastic modulus. However, values of such properties may differ at the nanoscale from the bulk. Prior studies of confined polymers have revealed that thin films exhibit $T_g$ values that are different from the bulk, and that this phenomenon was dependent on interfacial effects, i.e., the surrounding environment. Keddie et al. reported a $T_g$ reduction of polystyrene (PS) thin films supported on silicon substrates with decreasing film thickness via ellipsometry[11–13]. In contrast, they reported a $T_g$ increase for poly(methyl methacrylate) (PMMA) thin films when supported on the same substrate with decreasing film thickness. Since these original investigations, the influence of confinement and interfacial effects on $T_g$ of polymer thin films have been confirmed by various methods including fluorescence spectroscopy[14] dielectric relaxation spectroscopy (DRS)[15], dynamic mechanical analysis (DMA)[16], and Brillouin light scattering (BLS)[17,18]. Of particular importance, there is now growing evidence for the existence of a mobile layer at the surface of thin films, which has a crucial role in the observed deviations in $T_g$ with confinement[14,19–23].

Motivated by the increased use of polymer nanoparticles in technology and the scientific pursuit of uncovering a generality of confinement effects on $T_g$, irrespective of sample geometry, there has been recent interest in using nanoparticles as model systems[24,25]. Ediger and coworkers[26], reported a broadening of $T_g$ and a suppression in heat capacity change, $\Delta C_p$, at the glass transition with decreasing diameter for PS NPs stabilized with surfactants when suspended in an aqueous environment. Priestley and coworkers[27], reported a reduction in $T_g$ of PS NPs with decreasing diameter. In their investigation, the nanoparticles where stabilized without the use of surfactants. Later, Zhu and coworkers[28] confirmed that the suppression in $T_g$ of PS NPs with decreasing diameter was dependent on the presence (or lack) of surfactants (and surfactant type) at the nanoparticle interface. These findings, corroborated by Christie et al.[29], supported the notion that interfacial effects strongly influenced the glassy dynamics of confined polymers, including for PS NPs.

The leading argument for the observed confinement effects on the $T_g$ of PS NPs is associated with the existence a polymer surface layer that is more mobile than the bulk-like core[26–28]. Current techniques, e.g., differential scanning calorimetry, are limited to measuring $T_g$ of samples containing millions of NPs and cannot resolve the $T_g$ of a single NP. Moreover, because of the difficulty in characterizing the $T_g$ individual polymer NPs, and its segmental dynamics, the mobility at the surface of NPs has yet to be demonstrated. Hence, there remains a challenge to develop a technique that is capable of measuring the $T_g$ of individual polymer NPs and to provide direct evidence of surface mobility for these confined systems.

Penciu et al.[30] and Kuok et al.[31] first applied BLS to measure the eigenfrequencies of giant micelles and silica NPs of submicrometer size. For NPs with a diameter <100 nm, Raman scattering and pump probe techniques could be applied, however, fewer modes are resolved[32–36]. Recently, particle vibration spectroscopy by means of BLS has been advanced to measure particle mechanical eigenfrequencies to determine the elastic properties of polymer NPs[37–39]. For homogeneous spherical particles of diameter $d$, the lowest frequency BLS active mode is the $(s,1,2)$ spheroidal (s) vibration. Its frequency is given by the Lamb expression

$$f(s,1,2) = A c_t d^{-1} \qquad (1)$$

where $A$ is a dimensionless quantity that depends on the ratio between the transverse $c_t$ and longitudinal $c_1$ sound velocities in the particle[39–41]. The first attempt was to use Eq. (1), together with a model for the calculation of the Brillouin intensity, for assigning the observed peaks to the modes of an isolated NP[39,42,43]. Later, it was recognized that particle–particle interactions in clusters of NPs could influence the eigenfrequencies of individual NPs and lead to a low frequency broad band, which was attributed to the propagation of longitudinal waves[44,45].

Here, in an effort to advance the current state-of-the-art, we demonstrate that BLS yields information on the elastic properties of individual NPs while also accounting for the presence of interparticle interactions, which leads to a blue-shift and splitting of the $(s,1,2)$ mode. With the aid of a finite element method (FEM) model of a crystal of interacting spheres, it is possible to consider the separate contributions of these two effects. Based on this approach, we use BLS to measure the $T_g$ of PS NPs and to specify the mobile layer driven NP interactions. The experiments allow us to observe the vibrational modes of cluster of polymer NPs whose frequencies changed depending on the interactions among the NPs[35,36,45–47]. The $T_g$ of the PS NPs is measured by the temperature dependence of the vibrational modes. The enhanced mobility of polymeric chains on the surface layer caused as sudden increase in interparticle adhesion identified by a blue-shift of the fundamental mode $(s,1,2)$ and the low frequency interaction-induced $(s,1,1)$ broad band. Consequently, we demonstrate that measuring the temperature dependence of the eigenfrequencies is a powerful tool for studying the glass-transition behavior of PS under three-dimensional confinement and provides strong evidence for the presence of mobile surface layer mobility for PS NPs. The method can be extended to any polymer based NP with submicrometer size.

## Results

**Particle vibration spectrum.** Figure 1a shows the room temperature BLS spectra of clusters of PS NPs with three different diameters, $d = 141$, 202, and 707 nm referred to as $PS_A$-141, $PS_B$-202, and $PS_A$-707, respectively. The subscript A and B indicate different emulsion polymerization conditions (A and B stand for sodium-4-vinylbenzylsulfonate and acrylic acid chemistry; see details in Methods). The BLS spectra are plotted as scattered light intensity $I$ against the scaled frequency $f \cdot d$, where $f$ refers to the frequency. This presentation accounts for the $f \propto d^{-1}$ dependence of the eigenfrequencies of an isolated NP and further provides an adequate comparison of the shapes of $(s,n,l)$ bands, where the label denotes the radial $(n)$ and angular $(l)$ dependence of the displacement for the spheroidal modes $(s)$. For $PS_A$-707, the $(s,1,2)$ mode is a single peak represented by a Lorentzian line shape. For the smaller NPs, $PS_A$, and $PS_B$, the $(s,1,2)$ peak clearly deviates from a single line shape which can be phenomenologically reproduced by two Lorentzian curves. The peak positions in Fig.1a are different in all spectra, indicating that the mechanical properties of the materials are different or that there is an effect of the interactions among particles, or both.

Figure 1b shows the scaled frequencies $(f \cdot d)$ as a function of $d$ of the experimental $(s,1,2)$ peaks, taken at the intensity maximum. Also included in Fig. 1b are results of PS NPs taken from previously reported data[45,48]. The data are grouped into three $f \cdot d$ regions. Sample $PS_B$-202, $PS_B$-190, and literature PS NPs all fall in the intermediate region, $f(s,1,2) \cdot d = 1030 \pm 30$ GHz nm (or m s$^{-1}$). If we substitute these values into Eq. (1) for a homogeneous spheres[48], the transverse sound velocity within these PS NPs is calculated to be $c_t = 120 \pm 35$ m s$^{-1}$, which agrees with the corresponding, $c_t = 1210 \pm 20$ m s$^{-1}$ value of the bulk PS[39]. In the case of $PS_A$-707 for which the $(s,1,2)$ peak is well represented

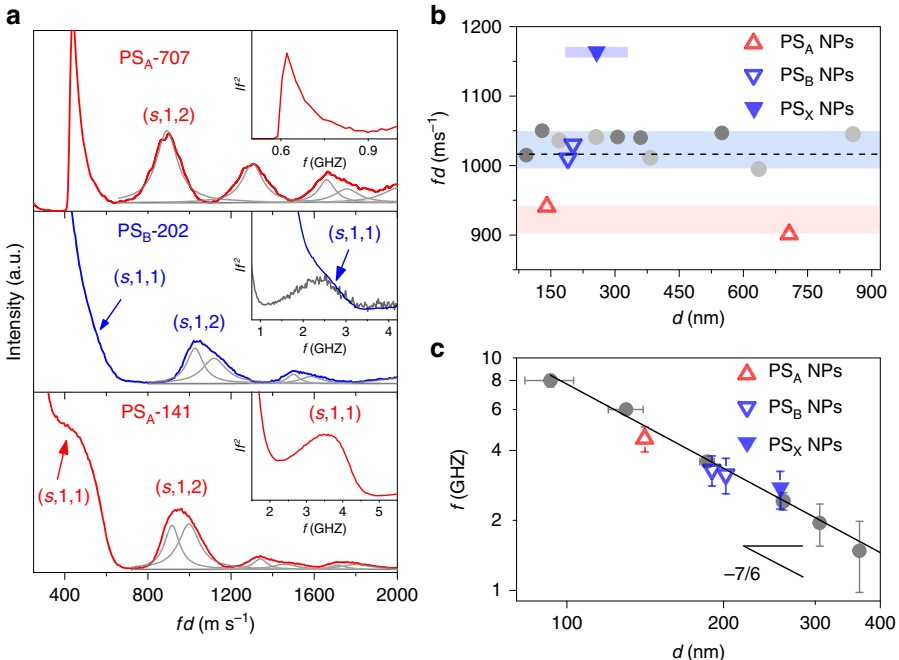

**Fig. 1** Size-dependent eigenfrequencies of nanoparticles. **a** Brillouin light scattering (BLS) spectra (*I* vs. *f* · *d*) for three PS NPs fcc clusters differing in the NP diameter: *d* = 141, 202, and 707 nm. The interaction-induced mode (*s*,1,1) (arrows) and the fundamental mode (*s*,1,2) are indicated in the plot. The gray solid lines denote the representation of the peak with either single or double Lorentzian lines. Inset: Reduced spectra $I \cdot f^2$ vs. *f* obtained from the spectra in the main plot. For PS$_B$-202, the reduced spectrum in black is recorded at high resolution. **b** $f(s,1,2) \cdot d$ vs. *d* and **c** $f(s,1,1)$ vs. *d* for PS$_A$ (red open triangles) and PS$_B$ (blue open inverse triangles) nanoparticles obtained from the cutoff frequency at 20% of the maximum intensity. Blue filled inverse triangles in (**b**) and **c** denote the crosslinked PS$_X$ NPs with *d* = 257 nm. The dashed line in (**b**) denotes the scaled frequency based on bulk PS elastic moduli. Gray and black filled circles in **b** and **c** are exported from refs.[45,52]. Blue and red filled areas in (**b**) indicate distinctive scaled $f(s,1,2)$ groups of PS$_A$ and PS$_B$, respectively. In **c**, linear gray line in indicates the scaling law $f(s,1,2) \propto d^{-7/6}$ derived by Johnson–Kendall–Roberts model. Inset schematically illustrates an interparticle adhesion activating the (*s*,1,1) mode. Horizontal and vertical error bars in (**c**) is obtained by the particle size distribution and from the method of determining the frequency at 20% of the peak intensity, respectively

by a single Lorentzian, $f(s,1,2) \cdot d = 900 \pm 10$ m s$^{-1}$, and hence $c_t = 1070$ m s$^{-1}$. Notably this value is about 10% lower than the $c_t = 1180 \pm 15$ m s$^{-1}$ measured in the corresponding contiguous film prepared by annealing the PS$_A$-707 NPs at 400 K. Therefore, the shear modulus $G = \rho c_t^2$ ($\rho$ refers to the mass density, 1050 kg m$^{-3}$ for PS) of the PS$_A$-707 NP is ~20% lower than that of the bulk.

For the PS$_A$-141, the frequency at the peak maximum leads to $f(s,1,2) \cdot d = 960 \pm 15$ m s$^{-1}$, which is slightly higher than the scaled frequency of PS$_A$-707. For both PS$_A$ particles, however, $f(s,1,2) \cdot d$ clearly falls below the value for all other PS NPs shown in Fig. 1b. For the latter, $c_t$ compares well with the bulk PS value also assumed by the contiguous films of all PS NPs. We attributed the reduced elastic modulus of both PS$_A$-141 and PS$_A$-707 colloidal film to the different emulsion polymerization procedures; PS$_B$-190 and PS$_B$-202 were synthesized using an acrylic acid comonomer (PS$_B$), whereas PS$_A$-141 and PS$_A$-707 were synthesized with sodium-4-vinylbenzylsulfonate (PS$_A$) (see Methods section). The scaled frequency of the two PS$_A$ NPs (inverse open blue triangles in Fig. 1b) is comparable to those of previously reported PS NPs because of the same chemical composition[45,48]. Importantly, this elasticity disparity implies geometry induced effects that are erased in the corresponding bulk films. This association to physical confinement is further supported by an additional NP, PS$_X$-257 formed with crosslinked (subscript X) polymer networks. The scaled $f(s,1,2)$ value marked at $f(s,1,2) \cdot d = 1200$ m s$^{-1}$ in Fig. 1b (filled inverse triangles), reflects larger elastic modulus than the corresponding contiguous bulk film (blue filled area in Fig.1b).

In Fig. 1a, nanocolloid films composed of smaller particles, PS$_A$-141 and PS$_B$-202 (and PS$_X$-257, which will be discussed later), show two major spectral differences compared to bigger PS$_A$-707: first, there is the appearance of the frequency peak below $f(s,1,2)$, better visible in the inset of Fig. 1a; and second peaks split (gray solid lines in Fig. 1a). Both effects are due to the adhesion among neighboring NPs. Figure 1c reports the $f(s,1,1)$ of the low frequency interaction-induced (*s*,1,1) mode for all PS NPs investigated. The reported $f(s,1,1)$ are those of the high-frequency cutoff in a $I \cdot f^2$ plot at 20% of the peak intensity owing to the nature of this interaction-induced mode[45]. These two differences are related to the sound propagation of longitudinal phonons in a cluster of NPs; hence, they will not be present if the interactions among the particles are switched-off. Its mode pattern corresponds to the pure translation of the sphere, i.e., the (*s*,1,1) mode with zero frequency in the free sphere[44]. All exchanged wave vectors **q**, with their magnitude $0 < q < q_{bs}$ (with the *bs* meaning back-scattering) contribute to this peak, which resembles the density of vibrational states (DOS) of the longitudinal branch (*s*,1,1,*l*). The reduced intensity $I \cdot f^2$ relates only to $\rho(f) \cdot C(f, \mathbf{q})$, where $\rho(f)$ is the DOS and $C(f, \mathbf{q})$ is the coupling coefficient that accounts for the BLS activity via the elimination of the thermal contribution $\frac{n(\omega,T)+1}{\omega} \sim \frac{1}{\omega^2}$ for $k_B T \gg \hbar\omega$ (where $n(\omega, T)$ is the Bose–Einstein factor) and $\omega = 2\pi f$. In addition, $I \cdot f^2$ allows a better resolution of the peak from the elastic scattering (Rayleigh line). The (*s*,1,1) mode was observed for the first time by Mattarelli et al.[45] for PS, and later reported for silica[35] and semiconductor[36] NPs. Longitudinal acoustic phonons of an fcc crystal of spherical particles have a cutoff frequency, $f(s,1,1) \sim$

$\sqrt{K_{eff}/M}$ governed by the effective constant $K_{eff}$ of spring between neighboring NPs of mass $M$. According to the Johnson–Kendall–Roberts (JKR) model[49], the scaling relation, $f(s,1,1) \propto d^{-7/6}$ (see Methods), well reproduces the experimental $f(s,1,1)$ for the present and reported PS NPs[45], as shown in Fig. 1c. In the case of $PS_A$-707, the $(s,1,1)$ mode is not resolved because of its low frequency (extrapolation of the scaling in Fig. 1c). Hence, the peak is obscured by the central Rayleigh peak (inset to Fig. 1a).

A comparison of the data in Fig. 1b, c shows that the frequencies of the $(s,1,1)$ and $(s,1,2)$ vibrations have different dependence on the strength of the interaction among the NPs and on the rigidity of the NPs. The latter has an important effect on the frequencies of the $(s,1,2)$ mode, which discriminates three groups for the $PS_A$, $PS_B$, and $PS_X$. The size of the NPs, which determines the strength of the interaction, has a minor role. On the contrary, the frequencies of the $(s,1,1)$ modes are governed by the strength of the interaction, $K_{eff} \propto d^{2/3}$. The fact that the NP vibration dynamics strongly depend on the particle mechanics, whereas the sound propagation in the colloidal film strongly depends on the strength of the interaction among the NPs, suggests a method for disentangling the consequences on the frequency shift of the $(s,1,2)$ mode of the isolated NPs. Mechanical softening of the PS NPs, with respect to bulk PS, will cause a red shift of the vibrational frequencies, while presence of interactions will produce a blue-shift of all frequencies compared to that of isolated NPs. The shift is accompanied by a broadening, since the interaction transforms the discrete spectrum of a free particle into phonon bands with finite dispersion.

These two effects are simulated by a simple FEM model of an fcc crystal of nearly spherical particles. The particles were obtained by cutting spheres to have a circle of radius $a_0$, as contact area among them. The strength of the interaction could be affected by changing $a_0$, as proposed by Saviot et al.[50] for the study of dimers. The interaction energy is proportional to the area and to the surface (adhesion) energy and produces a nearly constant attractive pressure on the contact area. At the equilibrium, these attractive forces are compensated by the repulsion due to the elastic deformation. Details are described in Methods (Theoretical model: phonons in a crystal of interacting spherical particles) section. Figure 2a shows the dispersion curves along the [100] direction of the lowest frequency phonons calculated for PS NP with $d = 141$ nm using $a_0 = 14$ nm, $c_l = 2350$ m s$^{-1}$, and $c_t = 1210$ m s$^{-1}$. We focused on the longitudinal $(s,1,1,l)$ and the five branches originating in the $(s,1,2)$ mode possessing significant BLS activity (red solid lines in Fig. 2a). The longitudinal and transverse sound velocities in the NP colloidal crystal are estimated from the low $q$ linear dispersion regime of the $(s,1,1)$ mode and are expectedly much lower ($c_l^{fcc} = 1270$ m s$^{-1}$ and $c_t^{fcc} = 810$ m s$^{-1}$) than in PS NPs[36,37]. For non-interacting NPs, $f(s,1,1)$ is zero (lower arrow on the right axis of Fig. 2a). The five $(s,1,2)$ branches (two degenerated) extend in a frequency region between about 7.4 to 9.1 GHz. All these frequencies are higher than $f(s,1,2) = 7.23$ GHz (upper arrow on the right axis of Fig. 2a), calculated for the free PS sphere using the same bulk $c_l$ and $c_t$ values. This occurs for all phonon bands since any constraint to the free motions occurring at the interface between the particles increases the vibrational frequencies.

Figure 2b illustrates the impact of interactions among NPs on the BLS active modes. For an isolated sphere ($a_0 = 0$) only a single Brillouin peak corresponding to the $(s,1,2)$ Lamb mode (Eq. 1) is present. Switching-on the interaction ($a_0 > 0$) results in two spectral features: (i) appearance of the interaction-induced peak $(s,1,1)$ which resembles DOS of propagating longitudinal phonons and (ii) blue-shift and splitting of the peak originating in

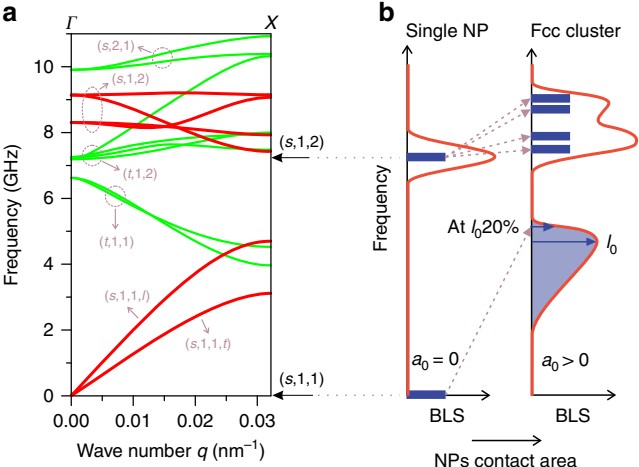

**Fig. 2** The effect of particle interaction on the eigenmode spectrum. **a** Calculated phonon dispersion (Γ-X) of an fcc cluster made of cut spheres with diameter $d = 141$ nm and contact area of radius $a_0 = 14$ nm, constituted by homogeneous polystyrene material ($c_l = 2350$ m s$^{-1}$ and $c_t = 1210$ m s$^{-1}$). The Brillouin active branches deriving from the $(s,1,1)$ and $(s,1,2)$ of the single sphere are marked with red solid lines. The weakly and non-active branches deriving from $(s,2,1)$ and the torsional modes, respectively, are indicated with green solid lines. The arrows on the right axis indicate the frequencies of the Brillouin active modes of the single particle. **b** Schematic diagram of Brillouin spectra for single particle ($a_0 = 0$) and fcc cluster ($a_0 > 0$), based on the dispersion in (**a**). The long and short blue arrows indicate the peak and the 20% pf the peak intensity positions

the $(s,1,2)$ mode. For the latter, the BLS spectrum suggests two Lorentzian peaks as shown with two experimental $(s,1,2)$ shapes of Fig. 1a. According to the BLS spectra, which can be calculated by the previous method[42] if the mode displacement field is available by FEM calculations (Supplementary Information), these two split peaks allows a rough estimation of the Lamb frequency, $f_L$, of a free sphere using $f_L = 2f_1 - f_2$, where $f_1$ and $f_2$ are the maximum frequencies of the two Lorentzians used to fit the experimental spectrum (rationalization of this relation is available in "Brillouin spectra of fcc crystals of spheres" section of the SI). As a result of this relation, $c_t$ in the individual NPs is calculated from Eq. (1). For $PS_A$-141 and $PS_A$-707, $c_t = 840$ ms$^{-1}$ whereas for $PS_B$-190 and $PS_B$-202, $c_t = 930$ ms$^{-1}$, which means that the effective $c_t$ does not depend on the particle size, but on the particle chemistry. The result of the calculation will be further discussed in the Discussion section to address the nanoconfinement effect.

**Temperature-dependent particle vibrational modes**. Well below $T_g$, temperature can have a minor impact in the frequency of the internal $(s,1,2)$ mode, since the thermal expansion coefficient is very small (for PS it is about $7 \times 10^{-5}$ K$^{-1}$)[51]. Thus, the observed red shift of the $f(s,1,2)$ mode reflects the decrease of $c_t(T)$[52,53]. Approaching the glass-transition temperature of the NPs, their $T_g$ is marked by the softening of $c_t$ and subsequent formation of a polymer film activated by chain mobility. Based on the preceding section, the interaction-induced $(s,1,1)$ mode associated with $K_{eff}$ and $M$ should be insensitive to temperature variation since the frozen segmental dynamics in the glassy state preclude change of interactions. Thus, an analysis of the temperature dependence of the $(s,1,1)$ and $(s,1,2)$ modes will allow not only to monitor and understand the thermal behavior of these polymer NPs, but also to identify thermal transitions. For simplicity, we first examine

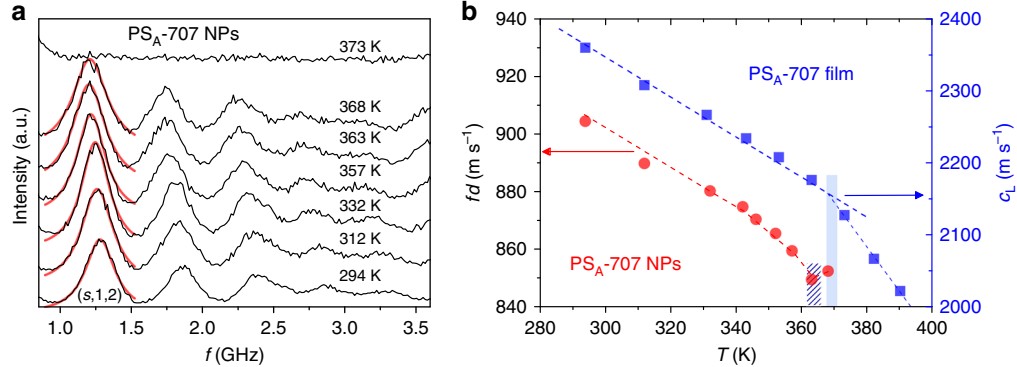

**Fig. 3** Temperature-dependent trend of eigenfrequencies. **a** Brillouin light scattering spectra (black line) of PS$_A$-707 at different temperature up to the formation of a contiguous polystyrene film with (s,1,2) mode peak fitted by a Lorentzian curve (red line). **b** The temperature dependent $f$(s,1,2). Red and blue circles represent the scaled frequency $f$(s,1,2)·$d$ for NPs (left $y$-axis) and the longitudinal sound velocity ($c_l$) of annealed bulk film (right $y$-axis), respectively. The hatched and filled areas in (**b**) indicate the softening $T_s$ and glass-transition $T_g$ temperatures, respectively

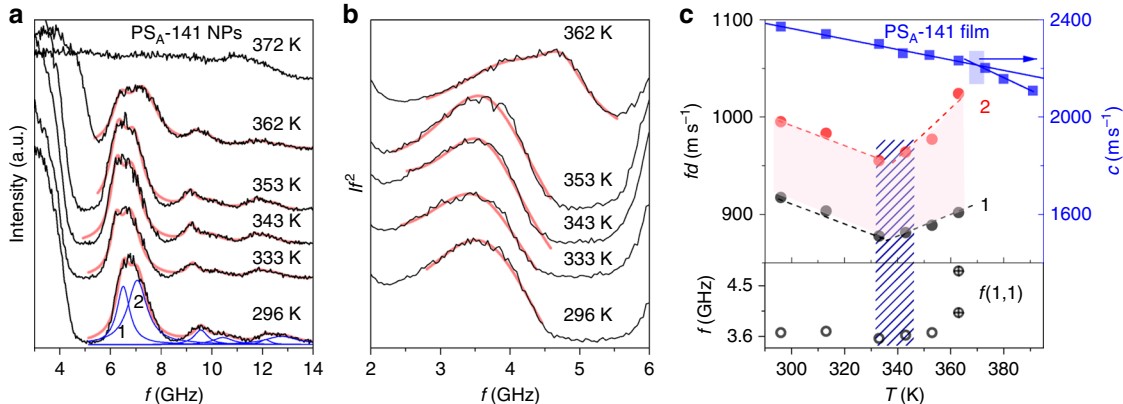

**Fig. 4** Influence of interparticle adhesion on eigenfrequencies and thermal transitions. Brillouin light scattering (BLS) spectra of PS$_A$-141 particles at different temperatures in two presentations: frequency $f$ vs. **a** intensity, $I$ and **b** $I$·$f^2$ (power spectra). Blue lines in (**a**) refer to Lorentzian lines of (s,1,2) mode. **c** Temperature dependence of the frequency, $f$(s,1,1) of the interaction-induced mode (low panel, opened circle) obtained at the maximum intensity and the scaled frequencies ($f$·$d$) of the split fundamental (s,1,2) mode (upper panel, black and red filled circles) of PS$_A$-141 nanoparticles. Two crossed open circles in the low panel indicate two Lorentzian curve fit of a $f$(s,1,1). The longitudinal sound velocity ($c_l$) recorded at $q = 0.0146$ nm$^{-1}$ for the contiguous PS film, obtained from the PS$_A$-141 particles annealed at 393 K, is also shown at right-hand side of $y$-axis (upper panel, blue solid square). For PS film measurement, the annealed PS film is heated from room temperature (296 K) to 393 K. The hatched and filled areas indicate the softening $T_s$ and glass-transition $T_g$ temperatures, respectively

PS$_A$-707 for which the (s,1,1) is not resolved and (s,1,2) can still well represented by a single Lorenzian line shape (Fig. 1a).

Figure 3a shows eigenfrequencies of PS$_A$-707 at different temperatures. The frequencies of the (s,1,2) modes are obtained from the location of the single peak (red solid line) at maximum intensity and are shown as a function of temperature in Fig. 3b. As the temperature is increased from 293 to 370 K, $f$(s,1,2) monotonically decreases reflecting the slowdown of $c_t(T) \propto f$ (s,1,2)·$d$ (red in Fig. 3b, left axis). Noticeably, this red shift is reversed into a blue-shift at about 363 K, which is here called the softening temperature $T_s$, and the eigenmodes vanish altogether at a higher temperature, $T_g \approx 372$ K, as seen in the featureless spectrum at 373 K in Fig. 3a. The disappearance of particle vibrational modes does not occur in either PS particles with a rigid silica shell or annealed bulk polymer films due to the rigidity of their structures[52,54]. In this respect, the disappearance of the vibrational modes in PS$_A$-707 at $T \approx 370$ K (Fig. 3b) is attributed to the structural deformation which occurs at their $T_g$; polymer flow induced by the activation chain dynamics (above $T_g$) prevents the emergence of eigenmodes due to formation of

contiguous PS films. Therefore, a temperature where eigenmodes disappear can be defined as $T_g$ of NPs. In Fig. 3b, the $T_g$ of the contiguous film estimated from the change of the slope, d$_{cl}$/d$T$, of the longitudinal sound velocity with temperature (blue filled squares) occurs at a very similar temperature with the NPs, $T \approx$ 369 K (blue filled area). The origin of the blue-shift of $f$(s,1,2) and the thermal transition at $T_s$ will be discussed below.

For the smaller particles, the $f$(s,1,1) mode is resolved from the central elastic scattering Rayleigh peak as shown in BLS spectra of PS$_A$-141 NPs in Fig. 4a. For a better resolution of the interaction-induced (s,1,1) mode, the data of Fig. 4a were scaled by multiplying the measured intensity by the square of frequency, $I$·$f^2$; see Fig. 4b. Several key features about the thermomechanical behavior and the adhesive interaction of PS NPs are revealed from Fig. 4a, b: (i) both the (s,1,1) and (s,1,2) modes exhibit an unexpected direction in frequency shifts; (ii) the peaks of both modes broaden and split into doublets as temperature increases; and, (iii) vibrational modes vanish at sufficiently high temperature as in the case of PS$_A$-707. In fact, only the (s,1,2) peak splits by the interactions, whereas the (s,1,1) assumes at the highest

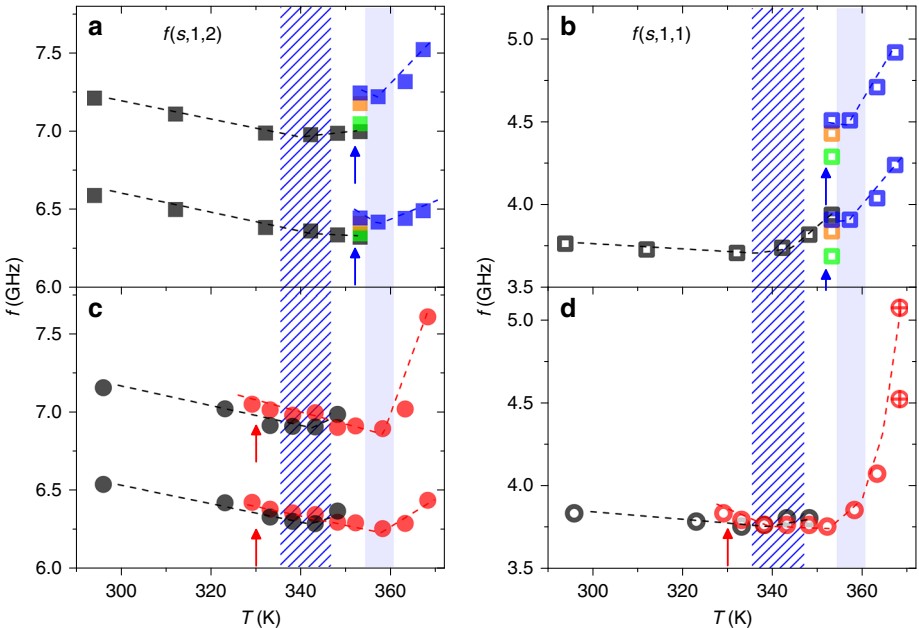

**Fig. 5** The nature of the softening transition temperature. The frequencies of the $(s,1,2)$ (**a**, **c**) and interaction-induced $(s,1,1)$ (**b**, **d**) modes obtained at the maximum intensity as a function of temperature. Black symbols indicate eigenfrequencies at the initial heating, and blue and red symbols are recorded from the second heating after particle annealing at 353 K (above $T_s$, blue arrow) (**a**, **b**), and at 326 K (below $T_s$, red arrow) (**c**, **d**), respectively. The frequencies of the two peaks in **a** and **c** are obtained from the Lorentzian representation of the $(s,1,2)$ mode. Annealing below $T_s$ causes apparent peak-splitting of the $(s,1,1)$ mode at 373 K (two crossed open red circles in **d**). Annealing above $T_s$ results in a doublet of $(s,1,1)$ mode at 353 K, and the hardening of both $(s,1,2)$ and $(s,1,1)$ modes is indicated by green (1 h), orange (9 h), and blue (18 h annealing) squares. In **b** and **d**, dashed and filled areas indicate the position of $T_s$ before and after the annealing process, respectively

temperature too broad band shape that can be represented by two Lorentzian curves. Their central frequencies are plotted in the lower level of Fig. 4c.

Interestingly, the shifts of the vibrational $(s,1,2)$ mode are not unidirectional, showing the $T_s$ of PS$_A$-141 similarly with bigger PS$_A$-707 particles. This behavior is demonstrated in Fig. 4c, where the upper graph contains the frequencies of the $(s,1,2)$ mode peak doublets in red and black and the lower plot depicts the frequency of $(s,1,1)$ mode. The junction of the red-shifted and blue-shifted regimes denotes a softening transition in the particle structure occurring below the $T_g$ at about $T_s = 340$ K. Given the phonon dispersion calculation described above, this inversion, as well as the increase in $f(s,1,1)$ at $T_s$, requires a significant increase in the contact area. Therefore, the blue-shift of the interaction-induced $(s,1,1)$ mode above $T_s$ indicates an increase in particle adhesion. For the $(s,1,2)$ mode, the blue-shift represents an increase in the contact area among particles resulting in the reinforcement of the interparticle adhesion force.

When particle interactions increase, the vibrational modes are predicted to split (Fig. 2), which is evident in Fig. 4a for the $(s,1,2)$. Above 343 K, the increase in particle interactions clearly raises $f(s,1,1)$ as shown in Fig. 4b, showing also strong blue-shift of the scaled $f(s,1,2)$ (Fig. 4c). As interparticle adhesion increases to the point that particles form a cohesive film, individual vibrational modes are abstracted and peaks vanish, which occurs at 372 K in Fig. 4a. This temperature is the $T_g$ for the NPs, and it closely mirrors that of annealed bulk PS for which the longitudinal sound velocity determined by BLS is plotted in Fig. 4c (see blue squares).

Despite the chemical similarity to PS$_A$-141, PS$_A$-707 has an overt softening temperature very close to $T_g$ (see Fig. 3b). Since the contact area is not linearly scaled by particle diameter ($a_0 \propto d^{2/3}$), PS$_A$-707 does not show a significant increase in the contact

area for $T_s < T_g$ (to compensate the $f(s,1,2) \propto d^{-1}$ dependence as shown in Supplementary Fig. 1). In contrast, PS$_A$-141 shows the $(s,1,1)$ mode and its blue-shift as temperature increases. Therefore, to overcome the red shift trend below the $T_s$, PS$_A$-707 requires a higher temperature to enhance the adhesion of the particles. Conclusively, $T_s$ of PS$_A$-707 is observed at a higher temperature than for PS$_A$-141. Because the mobile layer thickness is expected to be approximately independent of particle size, it constitutes a smaller portion of larger particles and contributes less to their mechanical properties of the overall colloidal film. Note that annealed bulk PS films lack a softening transition associated with a mobile surface layer.

PS$_A$-141 NPs are annealed below their glass-transition temperature to confirm the existence of the mobile surface layer. If present, the surface layer induces irreversible aging when NPs are annealed between their $T_s$ and $T_g$, but not in those annealed below $T_s$. The results of both experiments are plotted in Fig. 5; annealing is performed at 353 K, between $T_s$ and $T_g$, in Fig. 5a and b, but at 326 K, below $T_s$, in 5c and d. In Fig. 5a, b, the $f(s,1,1)$ and $f(s,1,2)$, respectively, are plotted for particles heated from 298 to 353 K in black symbols. The same softening temperature from Fig. 4 is present at 340 K, noted by the transition from a red to blue frequency shift. The particles are then annealed at 353 K for 18 h, and post annealing modes are plotted in blue. The increase in the $f(s,1,2)$ mode while annealing PS NPs at 353 K indicates increased particle adhesions resulted from the increase in contact area, which is absent in bulk PS[52]. This discrepancy between bulk polymer behavior and the colloidal polymer films suggests aging of the mobile surface layer.

The behavior of the $f(s,1,1)$ mode under thermal annealing is also interesting: it increases in frequency above $T_s$, but also broadens and appears as a doublet after annealing. Both indicate the increase in particle–particle adhesion, which can be attributed

to the fact that the mobile layer is softened so this increases a contact area abruptly at $T_s$ Finally, a new softening temperature of 360 K emerges during heating from 353 to 367 K due to the stiffening of the particles while PS NPs are annealed. These behavioral transitions that occur below the $T_g$, existence of a softening temperature, important blue-shift with change in shape of the (s,1,1) mode, and time-dependent vibrational modes, are unique to NPs and verify the presence of a mobile surface layer[55,56].

The results of Fig. 5a and b are corroborated by those presented in Fig. 5c and d, where particles are annealed below their $T_s$ at 326 K. Initially, particles are heated from 298 to 353 K as before, then slowly cooled to 329 K and annealed for 10 h. After annealing, the $f(s,1,1)$ and $f(s,1,2)$ show small increases in frequency due to increased particle aging from being heated temporarily above $T_s$ at the initial heating. Next, particles are heated from 329 to 367 K, which reveals the same increase in softening temperature to 360 K as in the high-temperature annealing scenario, because of the stiffened surface layer. However, the shape of the (s,1,1) mode does not appreciably change until heated to 373 K, demonstrating weaker adhesion among NPs. This can be attributed to incomplete softening of the surface layer during the first temperature scan and the thermal annealing. When particles are thermally heated longer to be completely transitioned at the second scan, the softened mobile layer increases the contact area, as well as the adhesion force, and this point is observed by the $f(s,1,1)$ doublet at $T = 373$ K. Despite the structural changes observed in the particles during annealing, the $T_g$ is consistently measured at 373 K. The results of Fig. 5 confirm the presence of a mobile surface layer and its significant influence on the interparticle adhesion among polymer NPs.

The effect of surface mobile layer can be augmented by chemically modifying NPs to enhance the elasticity and its contrast between the bulk-like core and mobile shell, respectively. Two particles were synthesized with this property, PS$_B$-202 and PS$_X$-257. PS$_B$-202 is a 202 nm PS particle copolymerized with acrylic acid, which increases the elastic modulus over bare PS particles. Previously, an increase in an elastic modulus of a PS$_B$-202 colloidal cluster had been shown with a larger scaled frequency $f(s,1,2) \cdot d$ than that of PS$_A$-141 in Fig. 1c. PS$_X$-257 is chemically similar to PS$_B$-202, but polymerized in the presence of a crosslinker at low concentration (5 wt.% divinylbenzene). As the particles grow, the crosslinker depletes, yielding particles with a crosslinked core and shell of free polymer chains[57]. The increased elastic modulus afforded by crosslinking is evident in the high-frequency of scaled $f(s,1,2) \cdot d$ presented in Fig. 6 over various temperatures; the BLS spectra of the three PS NPs particles (PS$_B$-190, PS$_B$-202, and PS$_X$-257) are shown in Supplementary Figs. 2, 3. Figure 6 shows that both PS$_B$-202 and PS$_X$-257 have softening temperatures around 353 K, higher than the 340 K of PS$_A$-141, which confirms their greater elastic modulus even in their surface mobile layer. The similar $T_s$ for PB$_B$-202 and PS$_X$-257 (Fig. 6) probably implies that crosslinking formation occurs in the core. As a result of the crosslinking the $T_g$ of PS$_X$-257 is higher than for PS$_B$-202 as indicated by the different temperatures at which contiguous films are formed in Fig. 6 (lower panel). The latter displays the same $T_g$, obtained from temperature-dependent sound velocity in the bulk films (Supplementary Fig. 4), with the $T_g$ of the corresponding NPs.

The effect of the surface layer mobility on particle mechanics becomes apparent when a rigid surface layer is added to PS NPs, shown in Fig. 6 (blue closed squares). PSS-437 NPs are crosslinked polystyrene particles confined by a thin (14 nm) hard silica shell with diameter $d = 437$ nm. The silica shell significantly retards the mobility of the surface polymer layer, which eliminates the softening temperature in turn[27]. Moreover, because

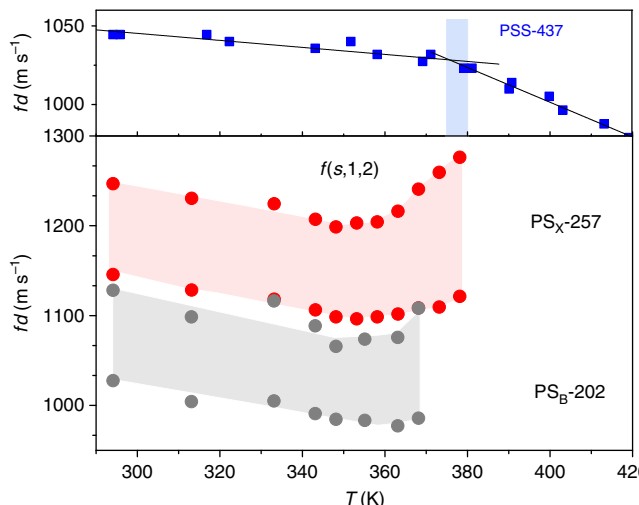

**Fig. 6** The influence of chemistry on the colloid vibration. The temperature dependence of $f(s,1,2) \cdot d$ of PS/silica core/shell (PSS-437) particle (blue, top) with diameter $d = 437$ nm, that is previously reported data[54], PS$_X$-257 (red), and PS$_B$-202 (gray). Blue filled area indicates the glass-transition temperature of PSS-437 nanoparticles. Red and gray areas highlight the gap between two Lorentzian peaks of (s,1,2) mode of PS$_X$-257 and PS$_B$-202, respectively

silica does not undergo a glass transition, the $T_g$ marks only the transition of the polymer core from glassy to liquid rather than the loss of vibrational modes as in all PS NP's. The glass-transition behavior of PSS-437 NPs resembles that of bulk polymer, as presented in Fig. 4c, but is about 10 K greater due to its crosslinked nature[58]. Consistently, higher $T_g$ was displayed by crosslinked PS$_X$-257 NP. The hard nanoconfined PSS-437 NP follows the thermal behavior of annealed PS film of PS$_X$-257 (Supplementary Fig. 4), which reflects the suppression of surface mobile layer by silica hard shell.

## Discussion

The particle vibration spectroscopy used here probes the mechanical properties of polymer NPs, varying in size and surface characteristics, including their glass-transition behavior and mechanical adhesion forces among NPs in a colloidal cluster. The temperature dependence of the vibrational modes yields information about the thermal relaxation of the NPs probed through the reinforcement of the interparticle mechanical adhesion induced by the presence of a surface mobile layer. FEM calculations of the phonon dispersion in the colloidal fcc cluster disentangle the effects of interaction from the free NP mechanics. Three findings emerge from this study; (i) robustness of $T_g$ to the sample geometry (Figs. 3, 4, 6), (ii) discovery of the softening transition below $T_g$ (Figs. 4, 6), and (iii) decrease in elastic (shear) modulus of NPs compared with their corresponding polymer films after thermal annealing (Fig. 1b).

To corroborate the first finding, we also measured $T_g$ by modulated differential scanning calorimetry (MDSC). The results are shown in Fig. 7a. For the examined samples, the $T_g$ measured by BLS (filled area in Fig. 7a) is about 7–10 K lower than that of MDSC (hatched area in Fig. 7a). This constant disparity in $T_g$ as measured by BLS and MDSC is attributed to two effects: (i) the different underlying physical mechanisms of each approach, whereby MDSC relies on the polymer segmental relaxation and BLS is based on the softening of the elastic modulus[53] and (ii) the quasi-isothermal vs. cooling ramp measurements of BLS and

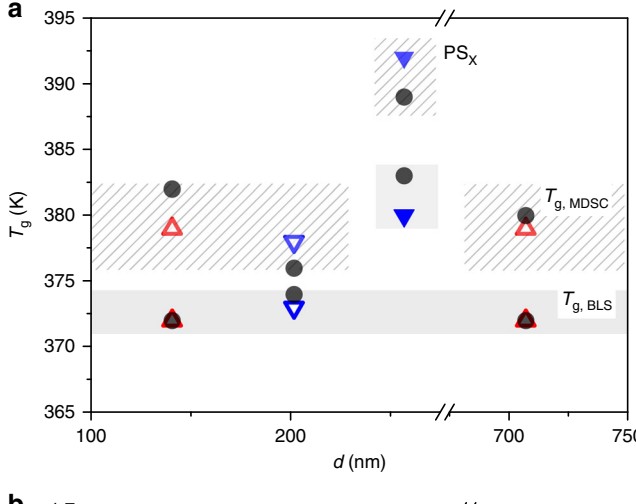

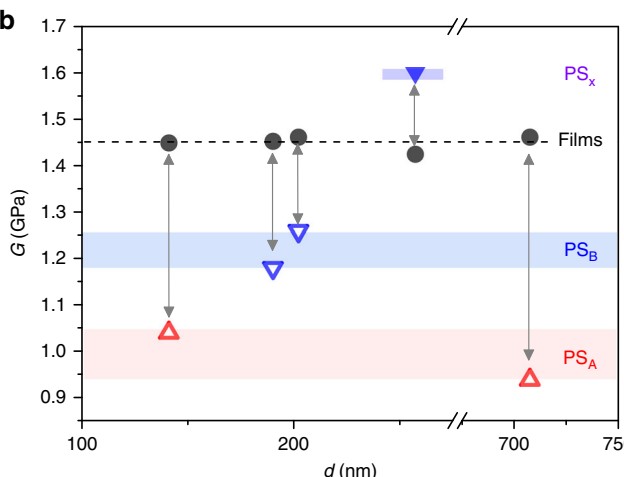

**Fig. 7** The glass-transition temperature and shear modulus of NPs and the annealed films. **a** The glass-transition temperature ($T_g$) both for nanoparticle powder state (red open triangles for $PS_A$, blue open inversed triangles for $PS_B$, and blue closed inversed triangles for $PS_X$) and for annealed bulk state (black closed circles) obtained by modulated differential scanning calorimetry (MDSC) thermograms (Supplementary Fig. 6) and Brillouin light scattering (BLS) indicated by the dashed and filled areas, respectively. The crosslinked $PS_X$-257 NP and corresponding annealed film assume higher $T_g$ values measured by the two techniques. **b** Shear modulus ($G$) of nanoparticles and the corresponding annealed films. The black dotted line indicates the average value of $G$ in the annealed bulk films. Arrows indicate nanoconfinement effect of NPs compared to their corresponding films. Blue and red areas refer to the softening of $PS_A$ and $PS_B$-type NPs, respectively. The purple area above indicates the hardening of crosslinked $PS_X$ NPs

MDSC, respectively[59]. Given the typical inconsistency between the two methods[52], the robustness of $T_g$ is verified, even if significant $T_g$ reduction is not captured. For $PS_A$ NPs, the addition of sodium-4-vinylbenzylsulfonate during synthesis might have an influence on the $T_g$ measurement, which is resulted from the different environmental factors. In case of $PS_B$ NPs, furthermore, the absence of the $T_g$ deviation can be related to the high concentration of acrylic acid, which could effectively behave as ionic surfactant[60]. Consequently, the additives alter the interfacial environment of the NPs as previously reported[28,29]. For instance, Feng et al.[28] illustrated a suppression and invariant in $T_g$ with confinement for PS nanoparticles without and with surfactants, respectively. Finally, we note that several studies have shown that

the magnitude of depression in $T_g$ of confined PS is dependent on cooling rate, with higher rates of cooling leading to a near complete elimination of size effects on $T_g$[61]. In this study, the rates of cooling/heating are well within range in which confinement strongly influences $T_g$. Hence, the absence of depression in $T_g$ of our nanoparticles cannot be related to rate effects in the measurement of $T_g$.

The second observation of the softening temperature provides strong evidence for the existence of a surface mobile layer atop the NP (qualitative calculation of the relation between contact area and $f(s,1,2)$ blue-shift above $T_s$ is available in Supplementary Information). However, in view of the absence of a depression in $T_g$ with decreasing NP diameter, this finding appears counter-intuitive and can suggest either decoupling between surface dynamics and $T_g$ with confinement[25] or the need of a sufficiently thick mobile surface to impact the $T_g$. The former possibility assumes that mobility is less affected by interfacial effects compared to $T_g$. The presence of sodium-4-vinylbenzylsoulfonate ($PS_A$) or acrylic acid ($PS_B$) in the surface layer effectively eliminates the $T_g$ confinement effect and promotes particle–particle adhesion with the presence of mobile surface layer. The temperature-dependent eigenfrequencies of core/shell NPs shown in Supplementary Fig. 5 confirm that the surface mobility is due not to a radial density gradient, but to a gradient in dynamics, and the effect of surface mobility on the softening temperature will be further investigated by modifying the surface layer of the NP in a core/shell topology.

Concerning to the third finding, Fig. 7b displays the elastic shear modulus for all examined NPs and the corresponding annealed films at ambient conditions. A comparison of the shear modulus for the two geometries (NP's vs. films) clearly reveals a large difference between the two. For both $PS_A$ and $PS_B$, the nanoparticle analog exhibits a much lower modulus in comparison to the annealed bulk films of each polymer. We suggest that this difference in modulus between NPs and annealed bulk films is a direct manifestation of the significant difference in the surface to volume ratio and surface mobility. This effect is consistent with reports of a dispersion in elastic modulus of thin PS films floated atop a liquid substrate[62]. The difference in the modulus between NPs of $PS_A$ and $PS_B$ is due to the different chemistries between $PS_A$ and $PS_B$. We also note that for each $PS_A$ and $PS_B$ NPs, in which only two different diameters were investigated, no clear trend in $T_g$ with decreasing NP diameter was observed; thought caution should be taken as only two NP diameters were available for investigation.

Finally, in the case of uncrosslinked ($PS_B$-202) and crosslinked ($PS_X$-257), the observed difference (~25%) in the $G$ values becomes negligible in the corresponding PS films. Mechanical hardening occurs only for the crosslinked NPs, whereas all uncrosslinked NPs are softer. The elasticity of the surface layer (uncrosslinked or crosslinked) can impact the NPs' $f(s,1,2)\cdot d$ (blue or red shift in Fig. 1b) depending on its shear modulus and thickness. During film formation (above $T_g$), structural rearrangement of the surface layer leads to contiguous films with all displaying bulk elasticity (Fig. 7b).

## Methods

**Polystyrene nanoparticle synthesis and characterization**. PS NPs were synthesized according to the procedure developed previously[63,64]. Briefly, $PS_A$ particles were synthesized in the presence of methanol and sodium-4-vinylbenzylsulfonate (NaVBS, Sigma-Aldrich) to control both the particle size and polydispersity. In 250 mL glass flask, a mixture of 130 mL of high-purity water (Millipore Direct-Q, resistivity ≥18.2 MΩ cm) and 40 mL of methanol (Fisher Scientific) was added under nitrogen condition. Next, 0.2 g of NaVBS (Sigma-Aldrich) was added. The solution was heated to 65 °C for 30 min, after which 25 mL of styrene monomer (0.5 wt.%) containing 0.5 wt.% potassium persulfate (Sigma-Aldrich) was added. The mixture was allowed to polymerize in excess of 8 h. $PS_B$ NPs are synthesized

with acrylic acid (AA, Sigma-Aldrich). A 230 mL of water was added into a glass flash equipped with a reflux condenser and the flask was heated to 80 °C under argon gas while stirring at 1200 rpm. After a stable temperature was reached, 3.2 g styrene and 0.2 g AA dissolved in 10 mL of water were added in sequence after 10 min equilibration time. The polymerization was initiated by adding 0.2 g of ammonium persulfate (APS, Sigma-Aldrich) dissolved in 10 mL of water. The stirring rate was kept at 700 rpm and the reaction was continued for 24 h. After the synthesis was complete, particles were washed three times by centrifuging and replacing supernatant water, and the particles were finally suspended in ultra-pure water. The crosslinked $PS_X$ particles were synthesized, as $PS_B$ NPs are prepared, in the presence of 0.16 g divinylbenzene (DVB, Sigma-Aldrich). NP sizes were measured using the fast Fourier transform (FFT) of scanning electron microscopy (JEOL JSM-7400F) images of well-ordered particle samples.

**Preparation of the colloidal film**. Glass substrates were cleaned in a sulfuric acid (Fisher Scientific) with 1 wt% of NOCHROMIX (Sigma-Aldrich) mixture for 8 h, then rinsed with water and dried under nitrogen gas. Before sample preparation, the glass substrates were cleaned under oxygen plasma. Colloidal films were prepared by drying a particle suspension between two glass substrates. A 200 μm spacer was placed between a glass slide (25 mm × 75 mm × 1 mm) and a coverslip (22 mm × 22 mm) with a 2 mm gap as a sample loading channel. Samples were injected into the channel and the medium was allowed to dry at room temperature. This procedure allows nanoparticles with different sizes to form long-range fcc lattice structures as shown in Supplementary Fig. 7a. Actually, the presence of a finite contact area reduces the distance between nearest particles ($a' = \sqrt{2(d^2 - 4a_0^2)}$) as reported in Supplementary Information, but the effect is very small. The packing fraction can be assumed 75% in all original samples. During the formation of the contiguous films the cluster should shrink.

**Modulated differential scanning calorimetry**. Modulated differential scanning calorimetry (MDSC) experiments were performed with polystyrene particles in suspension, as a dried colloidal powder, or an annealed bulk polymer state. Polymer particles were washed at least five times with water before preparing a sample. Suspensions used for MDSC consisted of 1 vol.% colloid. Dried powders were prepared by evaporating the aqueous medium under vacuum at room temperature. The annealed bulk polymer was prepared by annealing dried polymer NPs at 423 K for 24 h under vacuum. MDSC measurements were made with a Discovery DSC (TA Instruments) to measure the glass-transition temperature and all experiments were performed with a 4 °C cm$^{-1}$ heating rate and 30 s period of heat-only condition.

**Brillouin light scattering**. Brillouin light scattering (BLS) is a powerful and nondestructive technique useful for probing the thermal density fluctuations of a material by measuring the inelastic scattered light caused by thermally activated phonons. The scattering wave vector, $\pm\mathbf{q} = \mathbf{k}_s - \mathbf{k}_i$, is defined as the difference between scattered light $\mathbf{k}_s$ and incident light $\mathbf{k}_i$. In the transparent media, such as a polystyrene film, BLS spectra consist of a single doublet with a Doppler frequency shift with magnitude $f_{l,t} = \pm \frac{c_{l,t} q}{2\pi}$ at a given $q$, where $c_l$ or $c_t$ is the longitudinal or transverse wave velocity in the media, respectively. The wave vector $\mathbf{q}$ is independent on the refractive index in the transmission geometry and has a magnitude $q = \frac{4\pi}{\lambda}\sin\frac{\theta}{2}$, where $\lambda$ is the wavelength of incident light ($\lambda = 532$ nm) and $\theta$ is the scattering angle. In the turbid media, such as dry polystyrene particles, $q$ is ill-defined because of the strong multiple light scattering. Consequently, the BLS spectrum reveals the resonance modes of the colloidal particles that uniquely defined by the geometric and elastic characteristics of the particle cluster.

In temperature-controlled BLS experiments, temperature was monitored with platinum resistance temperature detectors and controlled within ±0.5 K of the set-point value with a custom temperature controller. Samples were isothermally equilibrated at each temperature for at least 20 min before recording a spectrum.

**Theoretical model of phonons in a crystal of interacting spherical particles**. For a free particle of spherical shape, the low frequency vibrations can be calculated in a continuous model as done by Lamb[40]. The modes are classified in torsional and spheroidal and described by ($p,n,l,m$). Torsional modes ($p = t$) have pure shear motions, whereas spheroidal modes ($p = s$) involve both shear and stretching. The $n$, $l$, and $m$ indices label the radial and angular dependence of the vibrational modes in spherical symmetry, in analogy to the atomic orbitals.

When the particles are deposited on a substrate, a more or less ordered cluster is formed, depending on the used technique. By depositing colloidal particles with narrow and sharp size distribution, large fcc crystals can be obtained. In the contact region, the spheres are deformed by attractive forces, even in the absence of external forces. According to the JFK model, for a small interaction, two spheres with diameter $d$ have a circular contact area of radius $a_0$ in case of no external force: $a_0 = \left(\frac{3\pi d^2 W_a}{8K}\right)^{1/3} \propto d^{2/3}$, where $W_a$ is the interaction energy per unit area of two flat parallel surfaces at a given distance and $K$ is the bulk elastic modulus. The interaction between two spheres can be described by a spring constant normal to

the contact surface given by

$$k_n = B W_a^{\frac{1}{3}} \left(\frac{d}{4}\right)^{\frac{2}{3}} K^{\frac{2}{3}} \propto d^{\frac{2}{3}} \tag{2}$$

The quantity $B$ depends on the model assumed for the distribution of the strains in the contact volume. Compressive strains are present in the central region, while tensile strains are present in the external region of the contact circle. Within the Johnson–Kendall–Roberts model,

$$B = \frac{9}{10}\left(\frac{3\pi}{2}\right)^{\frac{1}{3}} \cong 1.5 \tag{3}$$

In this approximation of small interaction, the vibrations of the cluster are a system of rigid spheres connected by springs. The system becomes a network of quasi-spheres, each one having a contact area of radius $a_0$ with the twelve surrounding spheres in the fcc crystal, as shown in Supplementary Fig. 8.

The sound propagates through the spheres maintaining nearly rigid structures with strains appearing only in a small region at the contacts. In an ideal fcc crystal, the maximum vibrational frequency, for waves propagating along the [100] directions, is

$$\omega(k_x) = 2\sqrt{2K_{eff}/M} \tag{4}$$

where $M$ is the mass of the sphere. The scaling laws with the particle size, i.e., $K_{eff} \propto k_n \propto d^{2/3}$ and $M \propto d^3$, produce another scaling relation for the frequency, $\omega(k_x) \propto d^{-7/6}$. In this regime, the longitudinal sound velocity of the system scales as $c_l \propto d\sqrt{(K_{eff}/M)} \propto d^{-1/6}$.

This motion can be described starting from the zero-frequency pure translation ($s$,1,1) mode as described in Fig. 2. There is another phonon band that derives from a zero-frequency mode of the free sphere: the ($t$,1,1) band in which the free sphere rotates freely and the spheres liberate in the cluster. This band has an important frequency dispersion. All other ($p,n,l,m$) modes form bands that, for small interaction among spheres, have small frequency shift and width change with respect to the relative discrete ($p,n,l,m$) mode of the free sphere, and the mode patterns within the spheres are nearly the same with small strains at the contact and phase changes that are governed by the $q$ vector.

The situation is very similar to that of a molecular crystal, where the sound propagation is determined by the interaction among molecules and optical phonon bands appear for each of the internal mode of the molecule. The symmetry is lowered from spherical to cubic at $q = 0$. The $2l + 1$ degenerate vibrations of the sphere split. For example, the $l = 2$ mode is split in a doubled and a triplet, the $l = 3$ in two triplets and a singlet. At $q \neq 0$, in general, no more degenerate modes are present and the $2l + 1$ components have all different frequencies. In an experiment at a defined $q$-value, a $q$-dependent spectrum of sharp lines will be present. In opaque samples, where multiple scattering occurs, all phonons contribute and the shapes of the ($n,l$) bands will resemble the shapes of the density of vibrational states of the crystal.

**Data availability**. The data that support the findings of this study are available from the corresponding author upon reasonable request.

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

## Acknowledgements

The work was supported by ERC SmartPhon (No. 694977). MDSC experiments were done in Advanced Materials Characterization Lab (AMCL, University of Delaware, Newark). B.G. acknowledges support from the Alexander von Humboldt foundation. R. D.P. acknowledges support from the Princeton Center for Complex Materials (PCCM), a

U.S. National Science Foundation Materials Research Science, and Engineering Center (Grant DMR-1420541). E.M.F. acknowledges support from the NASA (NNX16AD21G and NNX10AE44G) and the NSF (CBET-1637991).

## Author contributions

H.K., E.M.F., and G.F. designed the research; H.K. and E.K. synthesized particles and prepared samples; Y.C. and E.K. collected the experimental data; H.K., B.G., R.D.P., E.M.F., and G.F. analyzed data; B.G., M.S., and M.M. contributed to the theory and computer calculation; and H.K., B.G., R.D.P., E.M.F., and G.F. wrote the manuscript.

## Additional information

**Competing interests:** The authors declare no competing interests.

