## [Peer Review File · Nature Communications]

Reviewer #1 (Remarks to the Author):

This manuscript presents an extensive study of the vibrations of clusters of polystyrene (PS) nanoparticles and PS films using mainly Brillouin light scattering as a function of temperature. A careful analysis of the experimental spectra enabled the authors to disentangle the contributions of several parameters (radius of the nanoparticles, synthesis parameters, ...) and highlight the distinct role of the bulk of the nanoparticles and their surface. A very convincing link between surface mobility and the vibrations of clusters of PS nanoparticles is demonstrated.

This work builds on several previous works by some of the authors and others. As such, many of the experimental and numerical methods have already been presented before. The present manuscript makes an original use of them to address the issue of surface mobility in PS nanoparticles and films. The discussion and conclusions are supported by the data and very convincing. The tools are rather specialized and may be hard to grasp for newcomers in the field. But the topic is of interest for a large readership. For these reasons I believe this manuscript is suitable for publication in Nature Communications.

While the manuscript is generally well written, I think it must be improved in order to improve its clarity. Here is a list of issues which require modifications:

- The reason for the appearance of the $f(s,1,1)$ doublet is not clear. Fig.2b hints to a double peak structure in the vibrational density of states. However I do not understand why it does not appear at the same time as the doublet for $f(s,1,2)$. The authors should make it clear whether or not they can explain the appearance of this doublet.
- The authors choose to present PSA-707 before PSA-141 in Fig.3 and 4 and related text. The spectra for PSA-707 are easier to describe. But the existence of the softening temperature in these spectra is not clearly seen. In fact, I felt confident in the small increase of $f(s,1,2)$ at $T=368K$ only after having seen the results for PSA-141. Fig.4c clearly demonstrates the existence of the softening temperature T_S . Fig.3b pales in comparison. The blueshift is much more convincing in Fig.4c. For this reason, I believe the authors should consider to present PSA-141 (and Fig.4) first.
- In the discussion, the authors highlight three novel findings. The third one is "a confinement effect on the elastic (shear) modulus". I think the word "confinement" is not very appropriate in this case. This effect is primarily a difference between films and nanoparticles. No difference is reported between PSA-141 and PSA-707 despite the radii being quite different. The authors should justify the

use of the word confinement (previous use in the literature?) or switch to a more appropriate word (surface effect, dimensionality effect, ...).

- Does the observation of surface mobility in this work qualify as "direct" as written in the title? I suggest removing this word.

- Abstract: "we introduce a unique metrology of Brillouin light spectroscopy". Metrology is the science of weights and measures. I do not think it can be used in this context.

- How was the cutoff frequency at 20% determined for example for PSB-202 in Fig.1a? Did you fit both the tail of the Rayleigh peak and Brillouin peak? If so, what was the function used for the tail of the Rayleigh peak?

- It is very difficult to distinguish between $s(1,1,l)$ and $s(1,1,1)$. Maybe it will be easier in the final version. Otherwise using "L" instead of "l" (and "T" instead of "t") would help.

- Page 10: "At the equilibrium, these attractive forces are compensated by the repulsion due to the elastic deformation (Supporting Information)." Please write down where the reader should look in the Supporting Information.

Same remark page 12: "(rationalization of this relation is available in SI)"

- It is informative to see the featureless spectra at 372K in Fig.4a. Consider adding a similar spectrum in Fig.3a.

- caption of Fig.4: "The frequency of the longitudinal mode recorded at $q=0.0146 \text{ nm}^{-1}$ for the contiguous PS film obtained from the PSA-141 particles annealed at 393 K is also shown in (c)". The speed of sound is plotted, not the frequency.

- Fig.5: use 10 or 20K intervals between the ticks of the temperature axis and not 15K. It will make the scale much more useful. The other figures have a 20K interval between ticks.

- caption of Fig.5: "A gap between two Lorentzian fits of $f(s,1,2)$ doublets": Gap? Which gap?

- caption of Fig.5: "filled areas": did you mean "filled symbols"?

- The PSS-473 nanoparticles are incorrectly referred to as "PSS-437" and "PS@silica-437" in Fig.6 and its caption (twice).

Reviewer #2 (Remarks to the Author):

The manuscript by Kim et al. develops a new method based on BLS to study confinement and interfacial effects on PS nanoparticles made using three variable chemistries. The experiments are carefully performed and the method is described well and clearly in the paper. This method is quite unique in its ability to detect the onset of softening of NPs and the sound velocity inside the NPs, which can be used to measure the elastic modulus of the particles quite accurately.

However, I believe that the conclusions are a bit overstated and stronger than the data justifies. The authors list three new findings in the manuscript: 1- Robustness of T_g for sample geometry, 2- the discovery of the softening transition, and 3 confinement effects on the elastic modulus. Here are some of my thoughts about each conclusion and whether it is justified.

1- The authors don't measure any significant T_g reduction

as a function of NP size. However, they do find that the measured T_g using BLS is systematically lower than the T_g measured by DSC. They attribute these findings to different underlying mechanisms of the transition. While this is possible, the authors do not consider other potential effects. For example, it has been shown by various groups (Torkelson, Forrest, Fakhraai) that the magnitude of T_g may depend on the cooling rate. Given that the BLS experiments are quasi-isothermal and MDSC is performed at 4 K/min, it is quite possible that the difference is just due to the differences in rate.

Furthermore, since the T_g is not changing with size, these results do not provide any new unknown information about the NPs, which is somewhat disappointing, but not surprising given the size range investigated. I don't see this as a significant finding.

2- The softening transition is quite interesting. The authors attribute this to the enhanced mobility at the free surface. However, I caution that this may be different from the enhanced mobility on flat surfaces. At least for one of the systems studied here PSx the authors have good indication that the cross-linking density may be significantly different at the center and surface of the NP. Is it also possible that these NPs have a gradient in their density towards the free surface? How can this potential effect be investigated? Is it possible that the surface is softer because the density is lower? If so, that is not something that can be related to interfacial effects such as those observed in thin PS films, where the density appears to be quite uniform. This possibility needs to be discussed or ruled out.

Do the authors have a way to estimate the thickness of the soft layer or its contact area with the adjacent NPs? Is it possible that the surface is soft for all particles and only observed in thinner films because the contact area is larger for some reason? Do all films have the same packing fraction?

3- I think this last conclusion is the weakest. It is quite remarkable that the authors are able to measure the modulus in these NP systems. It is also great that they can distinguish changes in the elasticity as the chemistry of the sample changes, and that would have been a strong conclusion. The changes in elasticity, however, seem to only depend on the chemistry and not the NP size. As such, this cannot be called a confinement effect, where one would expect strong variations with particle size. If this is indeed a confinement effect what is the origin of the effect beyond the changes in the chemical composition of the system?

I could support the publication of this paper if the authors can address these questions or modify their conclusions to be in line with their findings. As such, I believe that major revisions are needed.

Reviewer #1 (Remarks to the Author):

This manuscript presents an extensive study of the vibrations of clusters of polystyrene (PS) nanoparticles and PS films using mainly Brillouin light scattering as a function of temperature. A careful analysis of the experimental spectra enabled the authors to disentangle the contributions of several parameters (radius of the nanoparticles, synthesis parameters, ...) and highlight the distinct role of the bulk of the nanoparticles and their surface. A very convincing link between surface mobility and the vibrations of clusters of PS nanoparticles is demonstrated.

This work builds on several previous works by some of the authors and others. As such, many of the experimental and numerical methods have already been presented before. The present manuscript makes an original use of them to address the issue of surface mobility in PS nanoparticles and films. The discussion and conclusions are supported by the data and very convincing. The tools are rather specialized and may be hard to grasp for newcomers in the field. But the topic is of interest for a large readership. For these reasons I believe this manuscript is suitable for publication in Nature Communications.

We thank the Reviewer for the encouraging opinion and the several pertinent comments and corrections.

Reviewer Comment

While the manuscript is generally well written, I think it must be improved in order to improve its clarity. Here is a list of issues which require modifications:

- The reason for the appearance of the $f(s,1,1)$ doublet is not clear. Fig.2b hints to a double peak structure in the vibrational density of states. However I do not understand why it does not appear at the same time as the doublet for $f(s,1,2)$. The authors should make it clear whether or not they can explain the appearance of this doublet.

Response: In fact, the presence of two peaks (s,1,1) in Fig.2b may transmit two different messages: (i) the lowest bar is the (s,1,1) transverse mode which is, however, not Brillouin active; (ii) the lowest bar could be a splitted longitudinal band, as for the (s,1,2), but it is not the case. Nevertheless, In order to remove this misunderstanding, we slightly modified Fig.2b (see below) in the revised Ms and explained the situation in the text.

- page 14, line 20, addition of the sentence: “In fact, only the (s,1,2) peak is split by the interactions, whereas at the highest temperature (s,1,1) assumes too broad shape that can be represented by two Lorentzian curves. Their central frequencies are plotted in the lower level of Fig.4c”
- Page 15, caption Fig.4, line 8, delete “split”.
- Page 16, line 5, modifying the sentence “When particle interactions increase, the vibrational modes are predicted to split (Fig. 2), which is evident in Fig. 4a for the (s,1,2). Above 343 K, the enhancement of the particle interactions clearly leads to increase of $f(s,1,1)$ as shown in Fig. 4b that also displays strong blue-shift of the scaled $f(s,1,2)$ (Fig. 4c).”
- Page 17, caption Fig. 5, line 8. ”Annealing below T_s causes apparent peak-splitting of the ...”
- Page 18, line 10. The behavior of the (s,1,1) mode under thermal annealing is also interesting: it increases in frequency above T_s , but also broadens and appears as a doublet after annealing.”
- Page 18, line 15. ”These behavioral transitions that occur below the T_g , existence of a softening temperature, important blue shift with change in shape of the (s,1,1) mode and ...”
- Page 18, line 23. “However, the shape of the (s,1,1) mode does not appreciably change until heated to 373K, demonstrating weaker adhesion among NPs”

Reviewer Comment

The authors choose to present PSA-707 before PSA-141 in Fig.3 and 4 and related text. The spectra for PSA-707 are easier to describe. But the existence of the softening temperature in these spectra is not clearly seen. In fact, I felt confident in the small increase of $f(s,1,2)$ at $T=368K$ only after having seen the results for PSA-141. Fig.4c clearly demonstrates the existence of the softening temperature T_s . Fig.3b pales in comparison. The blueshift is much more convincing in Fig.4c. For this reason, I believe the authors should consider to present PSA-141 (and Fig.4) first.

Response: We acknowledge what reviewer states and agree that it makes explicit that small particles resolve the issue at hand. In fact, while preparing this manuscript we have considered the Reviewer’s outline but we found much more difficult to start with inherently more complex case. Instead, presenting first the large NPs with a single (s,1,2) that displays a clear T_g and only a weak evidence of T_s becomes easier to contrast with the situation of the smaller NP displaying a clear T_s with "higher resolution" of modes.

Reviewer Comment

In the discussion, the authors highlight three novel findings. The third one is "a confinement effect on the elastic (shear) modulus". I think the word "confinement" is not very appropriate in this case. This effect is primarily a difference between films and nanoparticles. No difference is reported between PSA-141 and PSA-707 despite the radii being quite different. The authors should justify the use of the word confinement (previous use in the literature?) or switch to a more appropriate word (surface effect, dimensionality effect, ...).

Response: What we mean by “confinement effect” is their elastic modulus difference between NPs and their corresponding bulk film as shown in Figure 7(b). We already mentioned no size-effect between PS-141 and PS-707 in Results section so to more clearly describe this, we have revised this sentence with the following sentence in page 25, line 4:

For both PS_A and PS_B, the nanoparticle analog exhibits a much lower modulus in comparison to the annealed bulk films of each polymer. We suggest that this difference on modulus between NPs and annealed bulk films is a direct manifestation of the significant difference in the surface to volume ratio and surface mobility.

Reviewer Comment

Abstract: "we introduce a unique metrology of Brillouin light spectroscopy". Metrology is the science of weights and measures. I do not think it can be used in this context.

Response: We now introduce a unique “methodology” of Brillouin light spectroscopy

Reviewer Comment

Does the observation of surface mobility in this work qualify as "direct" as written in the title? I suggest removing this word.

Response: Since the softening transition is observed by the direct thermomechanical consequence of the presence of a mobile layer, it would be reasonable to still use "Direct" word in the title as it is.

Reviewer Comment

How was the cutoff frequency at 20% determined for example for PSB-202 in Fig.1a? Did you fit both the tail of the Rayleigh peak and Brillouin peak? If so, what was the function used for the tail of the Rayleigh peak?

Response: We have consistently used the frequency at 20% of the maximum of (s,1,1) peak in the reduced spectrum recorded under high resolution. For PS_B202 (Fig.1a) we have shown the reduced spectrum (blue) of the main plot recorded over broad frequency range and hence lower resolution. We have now included the reduced spectrum (black) recorded at higher resolution displaying a well defined (s,1,1) in Fig. 1a.

Reviewer Comment

It is very difficult to distinguish between $s(1,1,l)$ and $s(1,1,1)$. Maybe it will easier in the final version. Otherwise using "L" instead of "l" (and "T" instead of "t") would help.

Response: Using *l* italics (1,*l*) make it distinguishable. It only appears in Fig.2a.

Reviewer Comment

Page 10: "At the equilibrium, these attractive forces are compensated by the repulsion due to the elastic deformation (Supporting Information)." Please write down where the reader should look in the Supporting Information. Same remark page 12: "(rationalization of this relation is available in SI)"

Response: Page 10: (Supporting Information) indicates the details described in Methods (“Theoretical model: phonons in a crystal of interacting spherical particles” section).

This part is replaced by a sentence (in page 10, line 17): “Details are described in Methods (Theoretical model: phonons in a crystal of interacting spherical particles) section”.

Page 12: To specify where readers should look, the position of this information we have added in page 12, line 11 the following sentence: "rationalization of this relation is available in “Brillouin spectra of fcc crystals of spheres” section of the SI”

Reviewer Comment

It is informative to see the featureless spectra at 372K in Fig.4a. Consider adding a similar spectrum in Fig.3a.

Response: We have introduced the featureless spectrum at 373K in the revised Fig.3a and added a revised sentence in page 14 line 2: the eigenmodes vanish altogether at a higher temperature, $T_g \approx 372$ K, as seen in the featureless spectrum at 373 K in Fig. 3a.

Reviewer Comment

caption of Fig.4: "The frequency of the longitudinal mode recorded at $q=0.0146 \text{ nm}^{-1}$ for the contiguous PS film obtained from the PSA-141 particles annealed at 393 K is also shown in (c)". The speed of sound is plotted, not the frequency

Response: We have now corrected in page 15, line 8 by replacing this with the following sentence: The longitudinal sound velocity (c_l) recorded at $q=0.0146 \text{ nm}^{-1}$ for the contiguous PS film obtained from the PSA-141 particles annealed at 393 K is also shown at right-hand side of y-axis

Reviewer Comment

Fig.5: use 10 or 20K intervals between the ticks of the temperature axis and not 15K. It will make the scale much more useful. The other figures have a 20K interval between ticks

Response; Fig.5 is accordingly changed as shown below.

Reviewer Comment

caption of Fig.5: "A gap between two Lorentzian fits of $f(s,1,2)$ doublets": Gap? Which gap?

Response: "Gap" was used to indicate the difference between 1 and 2 peaks of (s,1,2) mode. We have now rephrased it in the caption (page 17, line 6): The frequencies of the two peaks in (a) and (c) are obtained from the Lorentzian representation of (s,1,2) mode.

Reviewer Comment

caption of Fig.5: "filled areas": did you mean "filled symbols"?

Response: Indeed, it was already corrected addressing the previous Reviewer's point

Reviewer Comment The PSS-473 nanoparticles are incorrectly referred to as "PSS-437" and "PS@silica-437" in Fig.6 and its caption (twice).

Response: We thank this Reviewer. We now consistently corrected the Fig. 6 (shown below), its caption and the text by only using PSS-437. (page 21, line 5)

Reviewer #2 (Remarks to the Author):

The manuscript by Kim et al. develops a new method based on BLS to study confinement and interfacial effects on PS nanoparticles made using three variable chemistries. The experiments are carefully performed and the method is described well and clearly in the paper. This method is quite unique in its ability to detect the onset of softening of NPs and the sound velocity inside the NPs, which can be used to measure the elastic modulus of the particles quite accurately.

However, I believe that the conclusions are a bit overstated and stronger than the data justifies. The authors list three new findings in the manuscript: 1- Robustness of T_g for sample geometry, 2- the discovery of the softening transition, and 3 confinement effects on the elastic modulus. Here are some of my thoughts about each conclusion and whether it is justified.

We thank the Reviewer for the positive comments on our article, and more specifically, for highlighting the “unique” method developed to monitor the dynamics of polymer nanoparticles. Below we provide detailed response to each point.

Reviewer Comment

1- The authors don't measure any significant T_g reduction as a function of NP size. However, they do find that the measured T_g using BLS is systematically lower than the T_g measured by DSC. They attribute these findings to different underlying mechanisms of the transition. While this is possible, the authors do not consider other potential effects. For example, it has been shown by various groups (Torkelson, Forrest, Fakhraai) that the magnitude of T_g may depend on the cooling rate. Given that the BLS experiments are quasi-isothermal and MDSC is performed at 4 K/min, it is quite possible that the difference is just due to the differences in rate.

Response: We agree with reviewer that cooling rate can have a significant impact on the T_g bulk and confined polymers; the latter being nicely demonstrated by Torkelson, Forrest, Fakhraai, etc. The heating rates for BLS and for DSC are about 1 K/min and 4 K/min, respectively. This modest difference in rate of measurement (irrespective of direction) – if both approaches were measuring the same physical manifestation of the glass transition – would lead to a lower T_g as measured by BLS in comparison to DSC, which is what is observed experimentally. This trend and magnitude is correct for the bulk samples, i.e., the contiguous films irrespectively of particle size as the memory of the NP geometry is lost. It also applies for particles with diameter greater than several hundred nanometers. As the diameter of the nanoparticles is reduced, the rate used to measure T_g in both approaches is held constant, and hence, the difference in rate is held constant. Because of this fact, and given the modest difference in rates, we do not believe that the absence in a measurable change in T_g with decreasing particle size can be attributed to a rate effect, i.e., rate is not changing at the different levels of confinement. Furthermore, in the original work on the influence of cooling rate on the T_g of thin films by Fakhraai and Forrest, higher cooling rates were needed to suppress the influence of thickness on T_g.

In response to this comment, we have made the following modification in the revised manuscript:

Page 22, line 6: This constant disparity in the T_g values as measured by BLS and MDSC is attributed to two effects: *i)* the different underlying physical mechanisms of each approach, whereby MDSC relies on the polymer segmental relaxation and BLS is based on the softening of the elastic modulus⁵³ and *ii)* the quasi-isothermal vs cooling ramp measurements of BLS and MDSC, respectively.⁵⁸

Page 22, line 18: Finally, we note that several studies have shown that the magnitude of depression in T_g of confined PS is dependent on cooling rate, with higher rates of cooling leading to a near complete elimination of size effects on T_g.⁶¹ In this study, the rates of cooling/heating are well within range in

which confinement strongly influences T_g . Hence, the absence of depression in T_g of the nanoparticles cannot be related to rate effects in the measurement of T_g .

Reviewer Comment:

Furthermore, since the T_g is not changing with size, these results do not provide any new unknown information about the NPs, which is somewhat disappointing, but not surprising given the size range investigated. I don't see this as a significant finding.

Response: Yes, most studies report that for bare PS nanoparticles the glass transition decreases with decreasing nanoparticle diameter, as noted in reference 25 and 27, 28, including in the size range investigated here. Hence, the robustness of T_g to sample geometry and size was unexpected. Therefore, providing a reason to the absence of a size effect on T_g , we believe, does provide new insights. The rationale for the lack of a T_g depression in PS NPs in this study is related to surface effects. While there are no surfactants, the high concentration of AA during synthesis leads to higher content of AA at the surface, which acts as an ionic surfactant. In ref. 28, both a suppression and invariant T_g was reported with decreasing diameter of PS nanoparticles with and without surfactants. This finding is in agreement with our study if we consider the degree which AA can concentrate at the surface of PS nanoparticles.

We have made the following modification to the manuscript in response to this comment:

Page 22, line 11: For PS_A NPs, the addition of sodium 4-vinylbenzylsulfonate during synthesis might have an influence on the T_g measurement, which is resulted from the different environmental factors. In case of PS_B NPs, furthermore, the absence of the T_g deviation can be related to the high concentration of acrylic acid, which could effectively behave as ionic surfactant.⁶⁰ Consequently, the additives alter the interfacial environment of the NPs as previously reported.^{28,29} For instance, Feng *et al.* illustrated a suppression and invariant in T_g with confinement for PS nanoparticles without and with surfactants, respectively.²⁸

Reviewer Comment:

2- The softening transition is quite interesting. The authors attribute this to the enhanced mobility at the free surface. However, I caution that this may be different from the enhanced mobility on flat surfaces. At least for one of the systems studied here PS_x the authors have good indication that the cross-linking density may be significantly different at the center and surface of the NP. Is it also possible that these NPs have a gradient in their density towards the free surface? How can this potential effect be investigated? Is it possible that the surface is softer because the density is lower? If so, that is not something that can be related to interfacial effects such as those observed in thin PS films, where the density appears to be quite uniform. This possibility needs to be discussed or ruled out.

Response: We thank the reviewer for this insightful comment. Note, that whether there is a gradient in crosslink density or not, the "new" method still provides for means to measure surface mobility, which is a focus of the article. We argue that the lack of T_g depression can be attributed to a higher presence of AA at the surface. Therefore, given the fact that the surface mobility is due to either a gradient in dynamics (at near constant density) or a gradient in density, there should not be a lower density at the free surface for the neat PS NPs, which do not contain crosslinked networks. To support the latter, we have investigated a core-shell particle (CS-168) consisting of a PS_A -141 core and a 14 nm shell. The shell is copolymer of styrene and trimethoxysilylpropylacrylate (TMSPA) with 10:1 styrene to TMSPA weight ratio. The density of this particle can be presumed to be very similar to PS. [ref. I. Tissot *et al.*, *Macromolecules* **34**, 5737-5739 (2001)]

Supplementary Figure 9. Glass transition temperature and softening temperature of core-shell NPs with PS_A-141 core and PS- like thin shell. (a) BLS spectra at different temperatures represented by Lorentzian curves and the featureless BLS spectrum at 380 K indicating the formation of a contiguous film; this temperature is higher than that measured for the PS_A-141 of Fig.4a. (b) Temperature dependence of the frequency, $f(s,1,1)$ of the interaction mode and the split (s,1,2) mode (lower panel) in CS-168 and PS_A-141 NP in an scaled version ($f(s,1,2) \cdot d$) (lower panel) and the frequency of the longitudinal phonon, f (at $q=0.0167 \text{ nm}^{-1}$) in the contiguous film obtained from the heating of CS-168 at 410 K (upper panel).

Based on Supplementary Figure 9, the CS-168 NP with homogeneously dense shell displays a higher T_s (by about 20K) and a higher T_g (by about 7K) than its parent PS_A-141 NP core. This corroborates the notion that the surface mobility is not the consequence of a gradient in density in the present polymer NP's. Instead, changing the surface dynamics in CS-168 NP impacts the softening transition temperature. Nevertheless, future work will systemically explore the role of surface capping and crosslinking on the observed properties of the nanoparticles.

In Page 23 line 10, the following sentences are added to describe this point:

The temperature dependent eigenfrequencies of core/shell NPs shown in Supplementary Figure 9 confirm that the surface mobility is due not to a radial density gradient, but to a gradient in dynamics, and the effect of surface mobility on the softening temperature will be further investigated by modifying the surface layer of the NP in a core/shell topology.

Also, in “Temperature dependent eigenfrequencies of core/shell nanoparticles” section of SI, Supplementary Figure 9 and its detailed descriptions above are included.

Reviewer Comment:

Do the authors have a way to estimate the thickness of the soft layer or its contact area with the adjacent NPs? Is it possible that the surface is soft for all particles and only observed in thinner films because the contact area is larger for some reason?

Response: The study reveals the presence of a surface mobile layer from the temperature dependence of both interaction (s,1,1) mode and the inherent (s,1,2) mode, which additionally is split due to the interactions. From the qualitative calculations, we can conclude that the observed frequency blue-shift above T_g results from the increase of the contact area radius, a_0 , of several nanometers. However, it is difficult to make any quantitative statement how this increase relates to the thickness of mobile layer.

Supplementary Figure 1. Graphical presentation of enhanced resolution with decreasing NP size. Reduced frequency $f(s,1,2) \cdot d$ as a function of the normalized contact area radius calculated at Γ point (Supplementary Figure 8). The two branches (upper triplet and lower doublet) originating in $(s,1,2)$ are presented. Horizontal arrows (long gray) depict the increase of the normalized contact area in the aggregation of the particles at low temperature, while the vertical arrows (green, shown only for the upper triplet) indicate the corresponding blue-shift. Increasing temperature above T_s (short gray arrow) leads to larger blue shifts (short green arrows) for smaller NP's (PS-141 vs PS-707)

Supplementary Figure 1 shows results of FEM calculations for fcc cluster of PS nanoparticles. This figure focuses only on the split and blue shift of the $(s,1,2)$ (calculated at $q=0 \text{ nm}^{-1}$) mode shown in a normalized presentation, $f(s,1,2) \cdot d$ vs a_0/d (normalized contact area radius). For any particle diameter, d , the increase of the particle-particle contact results in: (i) mode splitting and (ii) blue-shift. Nevertheless, the magnitude of these two effects depends on the particle size as we illustrate for $d=141 \text{ nm}$ (PS-141) and $d=707 \text{ nm}$ (PS-707). Following the JKR relation, $a_0 \propto d^{2/3}$ we take $a_0=14 \text{ nm}$ for PS-141 and $a_0=41 \text{ nm}$ for PS-707. (The numbers are arbitrary but provide the qualitative explanation.) Since $a_0/d \propto d^{-1/3}$, both (i) and (ii) interaction effects are larger for the smaller particle. This difference is even more apparent in the BLS original spectra (where the reduced frequency has to be divided by the particle diameter). Therefore, the mode splitting and the blue shift is less discernible for particles of bigger diameters and the temperature at which the mobile layer starts playing a role may change with surface to volume ratio of the particle. However, as the interactions give birth of $(s,1,1)$ mode, T_s is directly discernible in the temperature dependence of the $f(s,1,1)$.

Supplementary Figure 1 and the effect of contact area on the blue-shift of $f(s,1,2)$ described above is now included in the Supplementary Information with the following indication in the manuscript (page 23, line 2): qualitative calculation of the relation between contact area and $f(s,1,2)$ blue-shift above T_s is available in Supplementary Information.

Reviewer Comment:

Do all films have the same packing fraction?

Response: When NP suspensions are dried in the channel, NPs form fcc lattice structure. SEM images confirm long-range fcc structures for all samples. Actually, the presence of a finite contact area reduces the distance between nearest particles ($a'=\sqrt{2(d^2-4a_0^2)}$) as reported in SI), but the effect is very small. The packing fraction can be assumed 75% in all original samples. During film formation the cluster should shrink.

“Preparation of the Colloidal Film” of “Methods” section (Page 26, line 26): This procedure allows nanoparticles with different sizes to form long-range fcc lattice structures as shown in Supplementary Figure 5a. Actually, the presence of a finite contact area reduces the distance between nearest particles ($a'=\sqrt{2(d^2-4a_0^2)}$) as reported in SI), but the effect is very small. The packing fraction can be assumed 75% in all original samples. During the formation of the contiguous films the cluster should shrink.

Reviewer Comment:

3- *I think this last conclusion is the weakest. It is quite remarkable that the authors are able to measure the modulus in these NP systems. It is also great that they can distinguish changes in the elasticity as the chemistry of the sample changes, and that would have been a strong conclusion. The changes in elasticity, however, seem to only depend on the chemistry and not the NP size. As such, this cannot be called a confinement effect, where one would expect strong variations with particle size. If this is indeed a confinement effect what is the origin of the effect beyond the changes in the chemical composition of the system?*

Response: The reason we used “confinement” is based on the difference in elastic modulus between NPs and bulk polymer film. We compared NPs’ elastic moduli with those of their polymer films after thermal annealing corresponding NPs and what we found is regardless particle size and chemistry, bulk films have constant elasticity, whereas NPs show the softening of elasticity (crosslinked PS_x NPs have hardening effect). This lower elasticity can be attributed to surface mobility of nanoparticles. We agree that “confinement” is confusing to readers so to more systemically address this last conclusion, this part has been revised in page 25 by the following paragraph:

Concerning to the third finding, Fig. 7b displays the elastic shear modulus for all examined NPs and the corresponding annealed films at ambient conditions. A comparison of the shear modulus for the two geometries (NP’s vs. films) clearly reveals a large difference between the two. For both PS_A and PS_B, the nanoparticle analog exhibits a much lower modulus in comparison to the annealed bulk films of each polymer. We suggest that this difference in modulus between NPs and annealed bulk films is a direct manifestation of the significant difference in the surface to volume ratio and surface mobility. This effect is consistent with reports of a dispersion in elastic modulus of thin PS films floated atop a liquid substrate.⁶² The difference in the modulus between NPs of PS_A and PS_B is due to the different chemistries between PS_A and PS_B. We also note that for each PS_A and PS_B NPs, in which only two different diameters were investigated, no clear trend in T_g with decreasing NP diameter was observed; thought caution should be taken as only two NP diameters were available for investigation.

Also, this last conclusion iii) at “Discussion” section is modified by the following sentence in page 22, line 1:

iii) decrease in elastic (shear) modulus of NPs compared with their corresponding polymer films after thermal annealing (Fig. 1b).

Reviewer #1 (Remarks to the Author):

In this new version of the manuscript, the authors modified the presentation of the (s,1,1) Brillouin peak. They made it clear that it does not split the way the (s,1,2) peak does. They added a high resolution Brillouin spectrum in the insert of Fig 1(a). This was necessary to explain how the cutoff frequency at 20% of the maximum intensity was obtained. They also took into account the other minor corrections I spotted. I am still not convinced that the word "direct" in the title is appropriate but this is nitpicking.

I am satisfied with the corrections in the new version of the manuscript and the responses of the authors. I also believe that the changes implemented to answer the questions of reviewer #2 help improve the manuscript. Therefore, I recommend to accept this manuscript for publication in Nature Communications.

Some additional minor corrections are required for the caption of Fig 1 (page 7):

- line 14: "Blue filled inverse circles" :

- filled -> filled

- circles -> triangles

- line 16: "closed circles"?

- closed -> filled

Reviewer #2 (Remarks to the Author):

The authors have responded to all of my concerns and questions. I think the manuscript is good for publication.

Reviewer #1 (Remarks to the Author):

In this new version of the manuscript, the authors modified the presentation of the (s,1,1) Brillouin peak. They made it clear that it does not split the way the (s,1,2) peak does. They added a high resolution Brillouin spectrum in the insert of Fig 1(a). This was necessary to explain how the cutoff frequency at 20% of the maximum intensity was obtained. They also took into account the other minor corrections I spotted. I am still not convinced that the word "direct" in the title is appropriate but this is nitpicking.

I am satisfied with the corrections in the new version of the manuscript and the responses of the authors. I also believe that the changes implemented to answer the questions of reviewer #2 help improve the manuscript. Therefore, I recommend to accept this manuscript for publication in Nature Communications.

Some additional minor corrections are required for the caption of Fig 1 (page 7):

- line 14: "Blue filed inverse circles" :*
- filed -> filled*
- circles -> triangles*
- line 16: "closed circles"?*
- closed -> filled*

We thank Reviewer #1 for all constructive comments in the original but also the aforementioned corrections (Fig. 1) in the revised manuscript. We are now trust the clarity of our revised Ms has been significantly improved.

Reviewer #2 (Remarks to the Author):

The authors have responded to all of my concerns and questions. I think the manuscript is good for publication.

We are pleased that Reviewer #2 found convincing our thorough response.